# Pseudotime Analysis Reveals Exponential Trends in DNA Methylation Aging with Mortality Associated Timescales

**DOI:** 10.3390/cells11050767

**Published:** 2022-02-22

**Authors:** Kalsuda Lapborisuth, Colin Farrell, Matteo Pellegrini

**Affiliations:** Department of Molecular, Cell and Developmental Biology, University of California, Los Angeles, CA 90095, USA; klapborisuth@mednet.ucla.edu (K.L.); colinpatfarrell@ucla.edu (C.F.)

**Keywords:** pseudotime analysis, trajectory inference, epigenetic aging, DNA methylation

## Abstract

The epigenetic trajectory of DNA methylation profiles has a nonlinear relationship with time, reflecting rapid changes in DNA methylation early in life that progressively slow with age. In this study, we use pseudotime analysis to determine the functional form of these trajectories. Unlike epigenetic clocks that constrain the functional form of methylation changes with time, pseudotime analysis orders samples along a path, based on similarities in a latent dimension, to provide an unbiased trajectory. We show that pseudotime analysis can be applied to DNA methylation in human blood and brain tissue and find that it is highly correlated with the epigenetic states described by the Epigenetic Pacemaker. Moreover, we show that the pseudotime trajectory can be modeled with respect to time, using a sum of two exponentials, with coefficients that are close to the timescales of human age-associated mortality. Thus, for the first time, we can identify age-associated molecular changes that appear to track the exponential dynamics of mortality risk.

## 1. Introduction

Genome-based studies have provided insights into how DNA regulates biological processes involved in human aging [1,2]. However, DNA is largely invariant during the lifespan of an organism and genetic data provide limited insights on age-associated molecular changes [3]. By contrast, methylation of genomic DNA at cytosine bases is dynamic during a lifespan and is associated with gene regulation and human development [3,4,5,6]. Systematic DNA methylation changes proceed predictably with chronological age [7], which has led to the development of various approaches to model biological aging with DNA methylation data [8,9,10,11,12,13].

The first generation of DNA methylation-based aging models, known as epigenetic clocks, were devised to accurately predict chronological age [8,10,13]. The residual term of epigenetic clocks [8,10] has been used to measure biological age acceleration [14] and has been associated with mortality risk [15,16], along with various disease outcomes [17,18,19,20]. Since epigenetic clocks are typically fit by minimizing the difference between observed and predicted ages (DNAm age), such an approach enforces linearity between the two measures [8,9,21,22]. Although some epigenetic clocks, such as Horvath’s clock, attempt to model age-associated methylation changes in children as a specific nonlinear function of time [10,23], the nonlinear trends captured by these clocks may be biased due to a priori assumptions about the functional relationship between epigenetic changes and age [24,25].

However, despite chronological age being the strongest factor that drives age-associated traits, individuals with the same chronological age often experience varying age-associated health outcomes and mortality risk. To capture the age-associated biological variation more directly, recent DNA methylation-based aging models, such as DNAmPheno age, are also trained to estimate an aging biomarker, based on a combination of age-related phenotypes, clinical measures, and chronological age [26]. Nonetheless, similar to the epigenetic clocks, this approach still generates DNA methylation-based aging estimates that are linearly correlated to chronological age in adults [26].

To enable the detection of nonlinearity in epigenetic aging models, the Epigenetic Pacemaker (EPM) was proposed as an alternative approach to studying epigenetic aging. The EPM estimates DNA methylation data by modeling changes at each methylation site with respect to a hidden variable, the epigenetic state, that is non-linearly associated with time [27]. Both the site-specific parameters as well as the epigenetic state of each individual in a dataset are estimated using an expectation-maximization algorithm that minimizes the error between the observed and predicted DNA methylation values [27]. The EPM approach reveals that DNA methylation patterns of age-associated sites across tissue types change rapidly early in life and then progressively slow as humans age [7]. Nonetheless, to arrive at the optimal solution for the EPM, the optimization process begins with equating the epigenetic state of each individual with their chronological age, which may introduce some biases. It is, therefore, of interest to develop an unbiased model of the age-associated dynamics of DNA methylation through the human aging process, without using chronological age or making a priori assumptions about the functional relationship between DNA methylation changes and the epigenetic age.

To this end, we propose the use of pseudotime analysis, also called trajectory inference (TI) methods, to model the trajectory of DNA methylation changes over time. Over 70 variations of pseudotime analysis methods have been proposed [28], typically for longitudinal studies of single-cell expression data [28,29]. Pseudotime analysis orders cells based on similarities in their expression patterns and constructs a path along with this ordering, to assign each cell a value of pseudotime, a latent (unobserved) dimension providing a quantitative measure of the biological progress, such as cell cycle, cell-type differentiation, or cellular activation [28,29,30].

Generally, pseudotime analyses consist of the following three main parts: dimensionality reduction, identification of a trajectory, and determination of pseudotime values along the trajectory [31,32]. A cell’s pseudotime is typically the distance along the trajectory between the cell and the origin of the trajectory [32]. Therefore, pseudotime can be understood as an increasing function of chronological time but is not necessarily linear to chronological time [29].

In this study, we demonstrate that pseudotime analysis is applicable for modeling and visualizing the trajectory of DNA methylation changes in blood and brain tissues throughout the human lifespan. Our results show that there is a significant correlation between the methylation pseudotime and the non-linear dynamics of epigenetic landscapes, captured by the Epigenetic Pacemaker Model [27]. Moreover, the methylation pseudotime trajectory is well described by the sum of two exponential trends across a lifespan and suggests that the timescale of the methylation pseudotime may be related to the human population mortality rate.

## 2. Materials and Methods

### 2.1. Methylation Array Processing

Illumina Infinium HumanMethylation450 BeadChip microarray data from whole blood and brain samples in the Gene Expression Omnibus (GEO) [33] with more than 50 samples that do not have missing methylation BeadChip array intensity data (IDAT) files, repeated measurements of the same samples, and data from cultured cells or cancerous tissues [24] were selected and processed using minfi [34] (v1.34.0). To mitigate variations between different experiments, whole blood samples were selected if their median methylation probe intensity was greater than 10.5 and the difference between the observed and expected median unmethylation probe intensity was less than 0.4 [24]. We generated a blood training dataset composed of the processed Illumina 450 k array data [35,36,37,38] (*n* = 1605, age = 0–94 years) and a testing dataset consisting of 18 processed Illumina 450 k array datasets [18,39,40,41,42,43,44,45,46,47,48,49,50,51,52,53,54,55] (*n* = 4519, age = 0–99 years). A dataset of brain methylation array samples [6] (*n* = 675, age = 0–97 years) was also used.

### 2.2. Pseudotime Analysis

The pseudotime trajectory was inferred by, first, reducing the dimensionality of the input methylation data matrix with principal component analysis using prcomp in stats R package [56]. Then, we use mclust, an R package for Gaussian Mixture Modelling (GMM) [57], to cluster our samples in the low-dimensional space. The number of clusters and the covariance parameter were selected based on the Bayesian Information Criterion (BIC) as implemented in mclust [57]. Lastly, the pseudotime value for each training sample was assigned by Slingshot [58]. Slingshot first constructs a minimum spanning tree (MST) on the cluster centers, then applies a simultaneous principal curves method to obtain smooth trajectories of the MST, and finally assigns the pseudotime value for each sample according to the distance from the start of the trajectory to where the sample is orthogonally projected onto the curve [58]. The inferred trajectory was used to predict pseudotime values for the validation and test data that were projected onto the same principal components as the training set. The pseudotime analysis was conducted with a fixed random seed.

The whole blood training samples were stratified and split according to age for pseudotime trajectory training (*n* = 1283) and validation (*n* = 322). For all samples, methylation values at each site were quantile normalized by probe type [10] using the median methylation value of the training blood samples to mitigate batch effects. For training, we selected age-associated methylation sites with the absolute value of the Pearson correlation coefficient (PCC) larger than 0.4 between age and methylation values of training samples (*n* = 8510). The trajectory was then used to predict the pseudotime of the test data.

A dataset of brain methylation array samples was used for both trajectory fitting and pseudotime prediction. The samples were stratified and randomly split according to age for pseudotime trajectory training (*n* = 138), validation (*n* = 70), and testing (*n* = 467). Age-associated methylation sites were selected such that the absolute value of PCC between age and methylation values of training samples at each site is larger than 0.70 (*n* = 9446).

### 2.3. Epigenetic Pacemaker (EPM)

The Epigenetic Pacemaker (EPM) measures the epigenetic state of each sample by modeling methylation levels at each CpG site as a linear function of the epigenetic state (i.e., epigenetic age) of an individual [27]. Specifically, a methylation value $m_{ij}$, where i denotes the methylation site and j the sample, is modeled as follows:(1)$$\hat{m_{ij}}=m_{i}^{0}+r_{i}s_{j}+\epsilon_{ij}$$
where$m_{i}^{0}$ is the initial methylation value,$r_{i}$ is the rate of change,$s_{j}$ is the epigenetic state,$\epsilon_{ij}$ is the normally distributed error term.

The EPM starts with an input matrix of methylation values and uses a fast conditional expectation-maximization algorithm to output the optimal values of $m_{i}^{0}$, $r_{i}$, and $s_{j}$ by minimizing the error between the predicted and the input methylation values across all methylation sites. Chronological age was used as the initial estimate for epigenetic states for each sample [27].

### 2.4. Determining the Functional Form of Methylation Pseudotime

To determine the functional form that best describes relationships between methylation pseudotime values and chronological age as well as epigenetic age, we find the best fitting trend line with five functional forms, as follows:(2)$$y=a+bx+cx^{2}$$
(3)$$y=a+blog\left(x\right)$$
(4)$$y=a+b\sqrt{x}$$
(5)$$y=a\left(1-e^{-bx}\right)$$
(6)$$y=a\left(\left(1+c\right)-e^{-bx}-ce^{-dx}\right)$$
wherey is methylation pseudotime value,x is chronological age,a, b, c, and d are coefficients.

We denote the functions as quadratic (2), logarithmic (3), square root (4), exponential (5), and the sum of two exponentials (6) respectively. Our best-fit criteria include Akaike information criterion (AIC), R-squared value (R2), and Root Mean Square Error (RMSE). The AIC value is calculated using the stats R package [56]. We also used the package to fit a linear function and compute the correlation between pseudotime values and epigenetic ages using Pearson’s correlation coefficient (PCC) [56].

### 2.5. Analysis Environment

Methylation data processing, site selection, and epigenetic state detection were carried out using Python (v.3.9.6) in JupyterNotebook [59] using numpy [60], scikit-learn [61], EpigeneticPacemaker [27], Pandas [62], and Joblib [63] packages. Trajectory inference and additional analysis were carried out with R (v.4.1.1) [56] in RStudio [64]. The mclust [57], slingshot [58], RColorBrewer [65], ggplot2 [66], viridis [67], splitTools [68], and nlsr [69] packages were used.

## 3. Results

### 3.1. Methylation Pseudotime Is Nonlinear across the Lifespan

#### 3.1.1. Methylation Pseudotime of Whole Blood Tissue

Pseudotime analysis clusters a low-dimensional representation (PC = 2) of training samples and constructs a pseudotime trajectory across the cluster centers (Figure 1a). The inferred trajectory was used to assign pseudotime values of each training (Figure A1a) and validation sample (Figure A1b) projected on the same low-dimensional space. The trained model performed well on the training (R2 = 1.00, RMSE = 0.0133) and validation (R2 = 0.996, RMSE = 0.236) datasets. Figure A2a,b shows plots of the assigned pseudotime values against the chronological age of the training and validation set respectively.

The pseudotime value was predicted for each test sample projected onto the reduced-dimensional space used to construct the trajectory (Figure 1b). Figure 1c shows that the relationship between the predicted pseudotime and chronological age is characterized by a positive nonlinear trend, with a rapid change in young samples that gradually slows in older samples.

#### 3.1.2. Methylation Pseudotime of Brain Tissue

The trajectory of methylation data from brain tissue samples was inferred using a low-dimensional representation (PC = 2) of training samples at age-associated sites (Figure 2a). We then used the trajectory to assign pseudotime values to training and validation samples projected onto the low-dimensional space, as shown in Figure A3a,b, respectively. The trajectory was able to predict the pseudotime value well on the training (R2 = 1.00, RMSE = 0.00939) and validation (R2 = 0.980, RMSE = 1.108) datasets. The plots of the assigned pseudotime values against chronological age are shown in Figure A4a for the training set and Figure A4b for the validation set.

The representations of samples in the same reduced-dimensional space as model training were used for pseudotime value prediction of the test set (Figure 2b). Similar to the pseudotime of blood samples, the predicted pseudotime of brain samples increases in young samples and decelerates as age increases (Figure 2c). However, we note that the rate of change of pseudotime at very early ages is much higher in brain samples than whole blood samples. Furthermore, it abruptly decreases at approximately 5 years old, unlike the gradual change observed in whole blood samples.

### 3.2. Methylation Pseudotime Is Linearly Correlated to Epigenetic States Estimated by the EPM

Since we have previously shown that the change of epigenetic state across the lifespan is characterized by a similar nonlinear trend to that of methylation pseudotime [7], we investigated the relationship between the two values. We fit cross-validated (cv = 5) EPM [27] models to each of the blood and brain tissue training methylation array data at the same selected age-associated sites used for inferring pseudotime trajectory. We then used the fitted EPM model to predict the epigenetic state for samples in the test set.

Figure 3 and Figure 4 illustrate a clear linear relationship between the pseudotime and the estimated epigenetic state of blood and brain samples respectively. The statistically significant PCC values ($r_{blood}$= 0.972, $P_{blood}$< 0.001; $r_{brain}$= 0.997, $P_{brain}$< 0.001) further point towards a strong linear correlation between the two values for both tissue types.

### 3.3. Sum of Two Exponentials Best Describe Methylation Pseudotime across the Lifespan

Since the relationship between pseudotime and chronological age, in both blood and brain tissue, are similarly characterized by an initial rapid change that slows as the sample age increases, we sought a function that best describes this nonlinear trend by considering the AIC, RMSE, and R2 value as measures of fit. Specifically, we estimated the coefficient and evaluated the fit of the quadratic (2), logarithmic (3), square root (4), exponential (5), and the sum of two exponential (6).

#### 3.3.1. Functional Form of Methylation Pseudotime of Whole Blood Tissue

All functional forms can describe the general trend of pseudotime across the entire lifespan reasonably well (Figure A5). The logarithmic, exponential, and sum of two exponential trend lines generate improved fits to the methylation pseudotime of young samples. Amongst all functional forms, the sum of two exponentials has the highest R-squared value, with the lowest AIC and RMSE values (Table 1). This suggests that the sum of two exponentials best describes methylation pseudotime of whole blood samples as a function of chronological age (Figure 1C).

#### 3.3.2. Functional Form of Methylation Pseudotime of Brain Tissue

The pseudotime of brain samples can only be described well across the entire lifespan with the logarithmic and sum of two exponential functions (Figure A6). The sum of two exponentials has the highest R-squared value, with the lowest AIC and RMSE values for all functions (Table 2). Similar to the result of whole blood tissue, this indicates that the sum of two exponentials function of chronological age best approximates methylation pseudotime of brain tissue samples across the lifespan (Figure 2C).

## 4. Discussion

Several studies proposed the epigenetic clocks as an approach to model biological age by predicting the age of an individual based on the changes of DNA methylation profile over time [8,9,13]. The residuals of these epigenetic clocks are often used as aging and health biomarkers [14,15,16,17,18,19,20]. By construction, the age estimates produced by this approach (DNAm age) are constrained to be linear with chronological age [8,9,21,22]. Some epigenetic clocks, such as Horvath’s clock, address this limitation by modeling the methylation dynamics of children with a nonlinear function of chronological age [10,25]. However, the methylation aging trend that they captured may be biased due to a priori assumptions about the functional form underlying the relationship between methylation and chronological age.

The Epigenetic Pacemaker (EPM) provides an alternative approach to model age-associated methylation changes that allows the detection of nonlinearity while avoiding the aforementioned biases, by directly modeling the methylation matrix as a time-dependent entity. The EPM shows that DNA methylation changes at age-associated sites are nonlinear with time [7]. However, the EPM approach uses the known age of each individual as the starting value of the optimization, which may generate some bias in the epigenetic state estimation. It is, therefore, of interest to also develop a model of the dynamics of DNA methylation changes through the human aging process that is completely independent of any assumption concerning the underlying relationship between the time-dependent methylation changes and the model estimates.

To accomplish this, we used pseudotime analysis to learn the trajectory of DNA methylation through the human aging process and to investigate the correlation between methylation pseudotime and the epigenetic state estimated by the EPM. Pseudotime analysis orders the samples based on their similarities, to assign pseudotime value as a measure of biological progress [28,29,30]. Unlike the epigenetic clocks and the EPM, pseudotime analysis does not require chronological age information or assume any specific functional form to track the methylation change.

By applying the pseudotime analysis to DNA methylation data, taken from a dataset of brain samples and 18 datasets of blood samples, we observe consistent trends for both datasets. Across both tissue types, the relationship between chronological age and the pseudotime values is characterized by a positive nonlinear trend, with a high rate of change in young samples that slows as sample ages increase. Since pseudotime is a measure of biological progression, the positive trend reflects the progression of each sample along the epigenetic aging process and cannot be understood simply as the increase in overall methylation level over time. The rate of methylation change captured with pseudotime also reflects the rapid physiological development in the early years of life that decelerates towards old age.

We observe that the methylation pseudotime and the epigenetic states estimated by the EPM have a statistically significant linear correlation. This suggests that the pseudotime of both blood and brain tissue closely tracks epigenetic states. Since methylation pseudotime values were estimated without using chronological age, its correlation to the epigenetic state also provides biological support for the nonlinear epigenetic aging trends first captured by the EPM.

However, unlike epigenetic clocks, pseudotime analysis, in its current form, can only track the dynamics of methylation change and is not an accurate age predictor. Moreover, we have not yet studied whether deviations from pseudotime trajectories can be understood as a biological age acceleration measure.

To quantitatively understand the timescales over which DNA methylation is changing across a lifespan, we fit the pseudotime trajectories to various analytical functions. We found that the observed nonlinear trends are best described by the sum of two exponentials. Since we are able to detect such trends across two different tissue types, the sum of two exponential functions may provide a consistent description of the human epigenetic aging trajectories. The sum of two exponential trends can be interpreted as describing two independent exponential processes that occur simultaneously, one of which is faster than the other. Age-related human mortality is characterized by the Gompertz equation [70], μ(x) = αe^βx, such that μ(x) is the mortality rate at age x and β describes how fast the mortality rate increases with time. The value, β, for the human population is found to be 0.08, which results in a doubling of mortality risk every 8.5 years [71,72]. For the whole blood dataset, we find that one of the two coefficients is 0.1, which results in a doubling time of 6.93 years. This coefficient is remarkably close to 0.08, the Gompertz coefficient. This suggests that the timescale of epigenetic changes in blood is similar to that observed in mortality risk, which has a doubling time of around ten years. By contrast, the slower timescale of the brain data is slightly slower than the Gompertz beta, at 0.04 instead of 0.08. The beta of the brain data results in a doubling time of 17.33 years instead of 8.5 years. This may indicate that brains age epigenetically at a slower rate than blood, although even the brain’s exponential timescale is not far off from that of the Gompertz. Moreover, the faster brain timescale we observe in our double exponential fit may reflect the very rapid changes that occur during brain development in the first 6 years of life [73].

What is the implication of the observation that DNA methylation patterns change exponentially at timescales similar to those observed in human mortality studies? One hypothesis is that the epigenome is set early in development, and then as we age, the methylation patterns randomly drift and become more diverse, due to epimutations and changes in immune cell type composition. This drift is characterized by an exponential process and, therefore, is asymptotically approaching a level of maximum diversity or entropy, at which point the organism has the maximum risk of dying. In other words, as we age, we are exponentially losing the epigenetic specification of our cells, and this process is tracking the exponentially increasing risk of mortality. In support of this hypothesis, we show that the timescales of these molecular changes are similar to the timescales of mortality risk.

In this paper, we demonstrate that pseudotime analysis can be applied to DNA methylation data to model the trajectory of its changes over time, without relying on a priori knowledge of the functional form of methylation changes with time. The strong correlation between methylation pseudotime and epigenetic states suggests that both approaches capture the nonlinear trends in the dynamics of epigenetics during a lifespan. Moreover, we provide evidence that the timescale for these processes is intriguingly similar to the timescale of mortality risk, further buttressing the hypothesis that aging is at least in part driven by epigenetic drift.

In future studies, we plan to apply pseudotime analysis to methylation data from tissues other than whole blood and brain, to examine whether methylation pseudotime is described by the sum of two exponentials and strongly correlates to the epigenetic states across a wider variety of tissue types. We will also investigate whether exponential methylation trends are observed in other species and track the mortality risk timescales for shorter-lived organisms.

## Figures and Tables

**Figure 1 cells-11-00767-f001:**
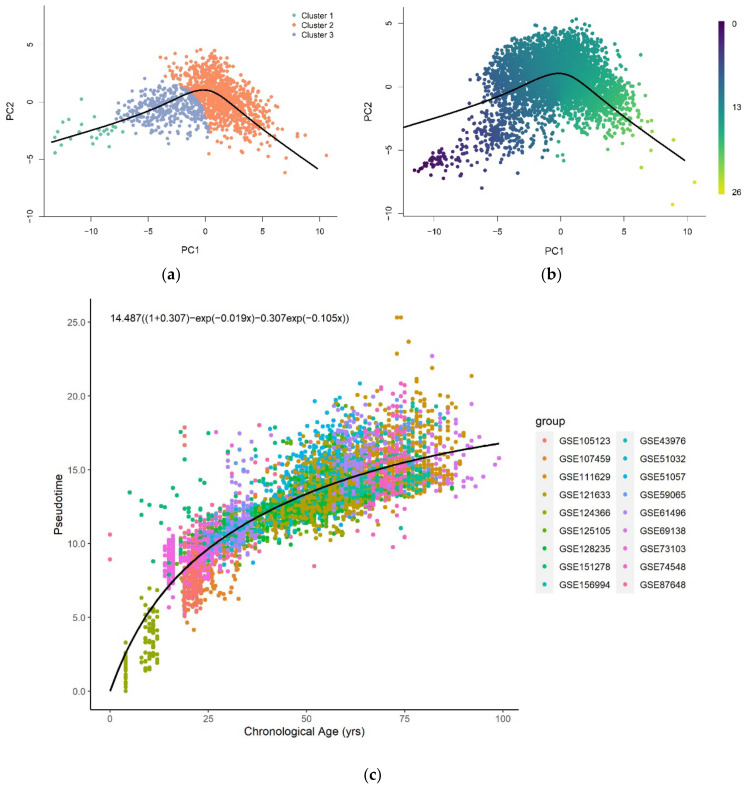
(**a**) The pseudotime trajectory inferred from clusters of training methylation array data of whole blood tissue in low dimensional space (PC = 2); (**b**) Methylation array data of whole blood tissue were projected onto low-dimensional space used for training. Each data point is colored according to the methylation pseudotime predicted with the pseudotime trajectory shown; (**c**) Methylation pseudotime of whole blood tissue samples.

**Figure 2 cells-11-00767-f002:**
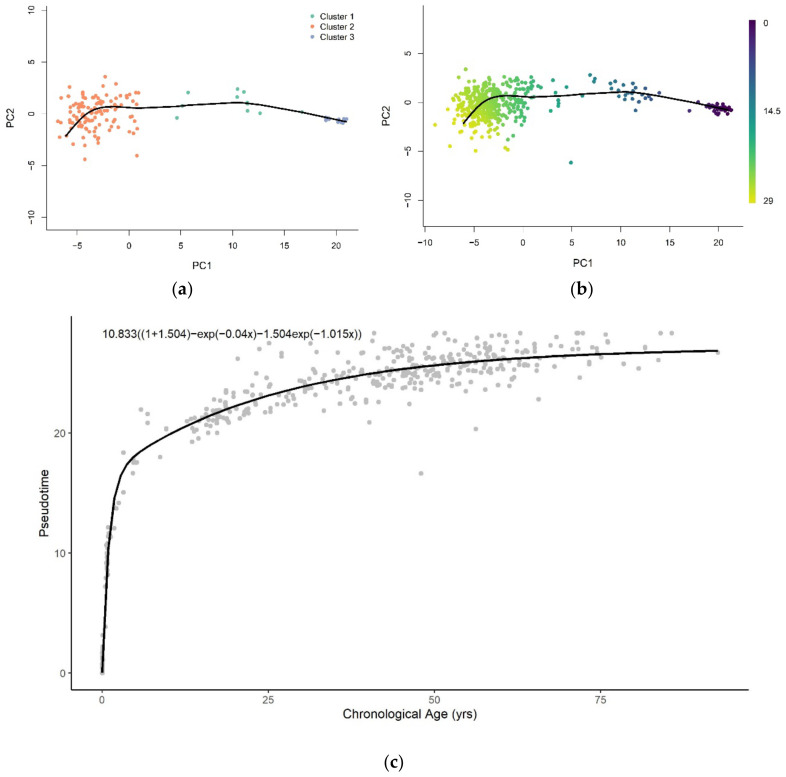
(**a**) The pseudotime trajectory inferred from clusters of training methylation array data of brain tissue in low-dimensional space (PC = 2); (**b**) Methylation array data of brain tissue were projected onto low-dimensional space used for training. Each data point is colored according to the methylation pseudotime predicted with the pseudotime trajectory shown; (**c**) Methylation pseudotime of brain tissue samples.

**Figure 3 cells-11-00767-f003:**
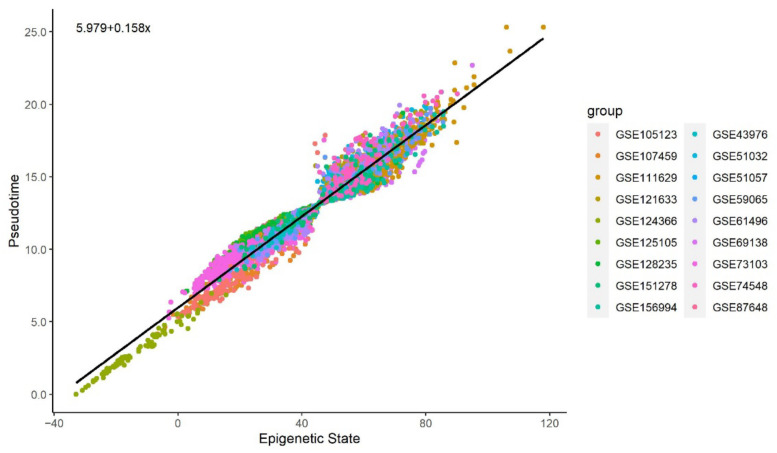
Methylation pseudotime vs. Epigenetic State of whole blood tissue samples.

**Figure 4 cells-11-00767-f004:**
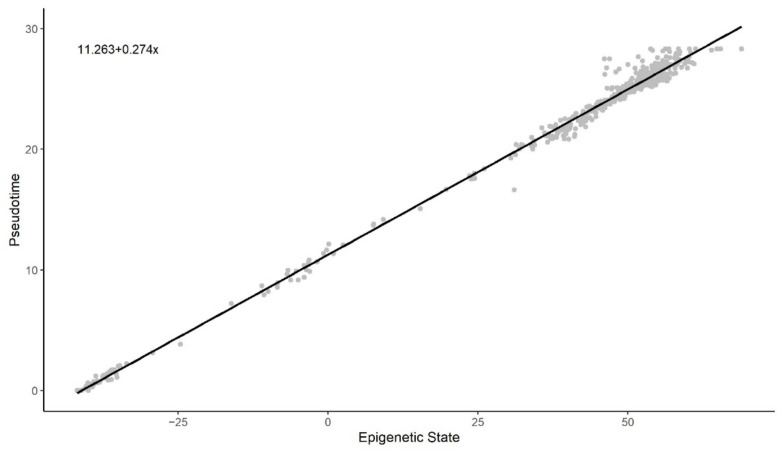
Methylation pseudotime vs. Epigenetic State of brain tissue samples.

**Table 1 cells-11-00767-t001:** Fit metrics of logarithmic, square root, exponential, and sum of two exponential functions to methylation pseudotime of whole blood samples.

Functions with Estimated Coefficients	AIC	RMSE	$R^{2}$
$y=-5.486+4.825\log\left(x\right)$	18,258.27	1.773	0.671
$y=1.23+1.686\sqrt{x}$	17,991.60	1.722	0.690
$y\;=3.56+0.277x-0.002x^{2}$	17,896.36	1.704	0.697
$y\;=\;16.531\left(1-e^{-0.035x}\right)$	17,913.38	1.708	0.695
$y=14.487\left(\left(1+0.307\right)-e^{-0.019x}-0.307e^{-0.105x}\right)$	17,826.79	1.690	0.700

**Table 2 cells-11-00767-t002:** Fit metrics of logarithmic, square root, exponential, and sum of two exponential functions to test samples of methylation pseudotime of brain samples.

Functions with Estimated Coefficients	AIC	RMSE	$R^{2}$
$y\;=11.512+3.572\log\left(x\right)$	1824.60	1.696	0.953
$y=6.100+2.802\sqrt{x}$	2422.50	3.217	0.831
$y\;=7.032+0.736x-0.007x^{2}$	2446.07	3.292	0.823
$y\;=\;24.738\left(1-e^{-0.456x}\right)$	2007.40	2.062	0.931
$y=10.833\left(\left(1+1.504\right)-e^{-0.04x}-1.504e^{-1.015x}\right)$	1555.68	1.266	0.974

## Data Availability

Publicly available datasets from the Gene Expression Omnibus (GEO) repository (https://www.ncbi.nlm.nih.gov/geo/ Last accessed: 14 October 2021) were used for analysis in this study. The datasets can be accessed with the following accession numbers: GSE42861, GSE43976, GSE51032, GSE51057, GSE59065, GSE61496, GSE69138, GSE73103, GSE74193, GSE74548, GSE87571, GSE87648, GSE97362, GSE105123, GSE107459, GSE111629, GSE121633, GSE124366, GSE125105, GSE128064, GSE128235, GSE151278, GSE156994.

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
