# Peer review of "Pseudotime Analysis Reveals Exponential Trends in DNA Methylation Aging with Mortality Associated Timescales"

_cells, 2022, doi:10.3390/cells11050767_

Round 1
Reviewer 1 Report
This group aimed to develop a model of the dynamics of DNA methylation that is independent of the relationship between time-dependent methylation changes with chronological age.
They used data from a DNA methylation dataset of brain and a dataset of blood samples processed with the Illumina Infinium HumanMethylation450 microarray.
To model DNA methylation dynamics they used trajectory inference.
Their analysis methodology is described in detail.
Using pseudotime analysis they found that there is a positive non-linear relationship between the predicted pseudotime and chronological age with a drastic change in young ages but slowing down in older age samples in both blood and brain samples.
Using correlation analysis with the Pearson's coefficient they found strong positive correlations between pseudotime and an epigenetic state estimate based on the previsouly descibed Epigenetic pacemaker model in both blood and brain samples.
They tested several functions and found that the sum of two exponentials is the function which best describes methylation pseudotime across lifespan.
They also provide evidence that the timescale of DNA methylation is similar to the timescale of mortality risk.
The findings are adequately and approrpiately discussed. Nevertheless, no limitations of the proposed modeling methodology are discussed. Data used to validate pseudotime analysis come only from blood and brain DNA methylation samples. Thus, the sum of two exponentials function has not been shown to describe DNA methylation pseudotime across lifespan in other tissue other than blood and brain. In addition, the correlation between pseudotime and estimated epigenetic state may not be significant in tissues other than blood and brain.
It must be discussed in the limitations section of the Discussion that the data used to validate pseudotime analysis come only from blood and brain samples.Thus, the sum of two exponentials function may not describe DNA methylation pseudotime across lifespan in tissues other than blood and brain.
In addition, the correlation between pseudotime and estimated epigenetic state may not be significant in tissues other than blood and brain.
Reviewer 2 Report
The manuscript named "Pseudotime analysis reveals exponential trends in DNA methylation aging with mortality associated timescales" of the authors' Lapborisuth et al., covers the trending topic of the epigenetic clock.
After reading the manuscript, I have several comments regarding the presented work that authors should address. Overall the manuscript has an informative, concise Introduction. However, chapter Results is not very convincing, rather confusing and too short. It is a mixture of statistical descriptions that should be moved to the Material and Methods and short statements about the increase or decrease of some parameters.
Unfortunately, the study's outcome is predictable - the increase of DNA methylation over time.
Authors should point and highlight the novelty and benefits of their approach/ work. How does the Epigenetics pacemaker compare to other models like DNAm age pr DNAm Pheno Age and what is his relationship regarding bivalent epigenetic marks such as H3K4me3 or H3K27me3?
The manuscript in the present form is not suitable for publishing and needs to be improved.
Round 2
Reviewer 2 Report
The manuscript named "Pseudotime analysis reveals exponential trends in DNA methylation aging with mortality associated timescales" of the authors' Lapborisuth et al., was extensively changed and improved according to the reviewer's suggestions. Authors responses to criticisms are appropriate, and the manuscript in the present form is suitable for publishing.